# Temporary Occlusion of Common Carotid Arteries Does Not Evoke Total Inhibition in the Activity of Corticospinal Tract Neurons in Experimental Conditions

**DOI:** 10.3390/biomedicines11051287

**Published:** 2023-04-27

**Authors:** Agnieszka Szymankiewicz-Szukała, Juliusz Huber, Piotr Czarnecki, Agnieszka Wiertel-Krawczuk, Mikołaj Dąbrowski

**Affiliations:** 1Department Pathophysiology of Locomotor Organs, Poznań University of Medical Sciences, 28 Czerwca 1956 r. Street, No. 135/147, 61-545 Poznań, Polandwiertelkrawczuk@ump.edu.pl (A.W.-K.); 2Department of Traumatology, Orthopaedics and Hand Surgery, Poznań University of Medical Sciences, 28 Czerwca 1956 r. Street, No. 135/147, 61-545 Poznań, Poland; pczarnecki@orsk.ump.edu.pl; 3Adult Spine Orthopaedics Department, Poznań University of Medical Sciences, 28 Czerwca 1956 r. Street, No. 135/147, 61-545 Poznań, Poland; mdabrowski@ump.edu.pl

**Keywords:** ischemic stroke, common carotid artery occlusion, corticospinal tract function, motor evoked potentials, temperature, rat

## Abstract

Temporary occlusion of the common cervical artery is the reason for ischemic stroke in 25% of patients. Little data is provided on its effects, especially regarding neurophysiological studies verifying the neural efferent transmission within fibers of the corticospinal tract in experimental conditions. Studies were performed on 42 male Wistar rats. In 10 rats, ischemic stroke was evoked by permanent occlusion of the right carotid artery (group A); in 11 rats, by its permanent bilateral occlusion (B); in 10 rats, by unilateral occlusion and releasing after 5 min (C); and in 11 rats, by bilateral occlusion and releasing after 5 min (D). Efferent transmission of the corticospinal tract was verified by motor evoked potential (MEP) recordings from the sciatic nerve after transcranial magnetic stimulation. MEPs amplitude and latency parameters, oral measurements of temperature, and verification of ischemic effects in brain slides stained with hematoxylin and eosin staining (H + E) were analyzed. In all groups of animals, the results showed that five minutes of uni- or bilateral occlusion of the common carotid artery led to alterations in brain blood circulation and evoked changes in MEP amplitude (by 23.2% on average) and latency parameters (by 0.7 ms on average), reflecting the partial inability of tract fibers to transmit neural impulses. These abnormalities were associated with a significant drop in the body temperature by 1.5 °C on average. Ten minutes occlusion in animals from groups A and B resulted in an MEP amplitude decrease by 41.6%, latency increase by 0.9 ms, and temperature decrease by 2.9 °C of the initial value. In animals from groups C and D, five minutes of recovery of arterial blood flow evoked stabilization of the MEP amplitude by 23.4%, latency by 0.5 ms, and temperature by 0.8 °C of the initial value. In histological studies, the results showed that ischemia was most prominent bilaterally in sensory and motor areas, mainly for the forelimb, rather than the hindlimb, innervation of the cortex, putamen and caudate nuclei, globulus pallidus, and areas adjacent to the fornix of the third ventricle. We found that the MEP amplitude parameter is more sensitive than the latency and temperature variability in monitoring the ischemia effects course following common carotid artery infarction, although all parameters are correlated with each other. Temporary five-minute lasting occlusion of common carotid arteries does not evoke total and permanent inhibition in the activity of corticospinal tract neurons in experimental conditions. The symptoms of rat brain infarction are much more optimistic than those described in patients after stroke, and require further comparison with the clinical observations.

## 1. Introduction

Ischemic stroke is a disease with a high mortality and an enormous socio-economic burden [1]. It is the second highest (87%) health problem in the world population, occurring more often than a hemorrhagic stroke (25%) [2]. The percentage of elderly patients with the incidence of ischemic stroke dramatically increases, but the average age of patients is decreasing [3]. Despite intensive experimental and clinical scientific efforts throughout recent decades, treatment options for acute ischemic stroke patients remain very limited [4]. Etiopathogenesis, the course and effects of ischemic stroke, seems to be the most important topic that impacts the best treatment strategies [5,6,7,8]. The cause of stroke is most often the aim of studies referring especially to people as well as the experimental studies on animals, and much is known on this topic [9]. However, effects of stroke before brain death seem to be unsatisfactorily described [10].

Among commonly used methods of clinical neurophysiology, such as electromyography and electroneurography, somatosensory (SEP) and motor evoked potentials (MEPs) were frequently chosen as precise methods in the evaluation of transmission in ascending and descending pathways in patients after stroke [11,12,13]. A number of studies in patients with ischemic stroke showed that MEP recordings are a sensitive prognostic indicator of the effects and orientation of treatment, aiming to restore the lost motor function [14,15,16]. MEPs recorded on the more paretic side were either absent or had low amplitude and prolonged duration, while the central efferent conduction time was only slightly increased [17,18,19]. Studies of MEPs in stroke patients provided descriptions of changes in their recording parameters up to 3 months after a middle carotid artery infarction [18].

Previous experimental studies with MEP recordings mainly from muscles, and less often from nerves, are aimed towards the description of ischemic stroke models following middle cerebral artery occlusion and include observations of changes in efferent transmission up to 6 h [20,21]. The reason this work has been undertaken is the phenomenon of variability in efferent transmission through the corticospinal tract as a result of occlusion in the common carotid artery, constituting the high incidence cause of ischemic stroke. Local carotid stenosis at 70% of its diameter occurs in 20% of stroke patients, but few papers clarified its effects [22].

Dougherty et al. [23] and Garcia [24] state that most rodents are the ideal models for observing the effects of stroke following selective carotid artery uni- or bilateral occlusion. The rat model commonly used for carotid artery occlusion is considered the most reliable in the evaluation of the effects of ischemic stroke on the function of the cortico-spinal system in experimental conditions [25], similarly, as applied in this study. Kety [26] believes that the blockage of the carotid artery does not cause significant changes in the cerebral circulation of most mammals, including humans, while Longa et al. [27] mention that the occlusion of intracranial carotid arterial branches in the rat does not cause stroke at all. In their opinion, this is due to numerous anastomoses forming connections between the branches of external and internal carotid arteries.

The available literature does not provide detailed descriptions of studies on the course of ischemic stroke in the rat after occlusion of the common carotid artery when transcranial magnetic field stimulation was used to assess its effects on cortical motor neuron excitability. Moreover, if such studies were undertaken, the recordings of evoked potentials from hindlimb muscles instead of nerves were performed. This last-mentioned site of recordings would increase the precision of measurements in the direct efferent transmission coming from the possible influence of anesthetics and changes in muscle motor end-plate function. Therefore, in this paper, an attempt was made to demonstrate the longest duration effect of cerebral ischemia in experimental conditions, which, despite its existence, is reversible in adverse effects on the activities of neurons in the motor cortex.

## 2. Materials and Methods

### 2.1. Animals and Study Design

Studies were performed on 46 adult male Wistar rats weighing from 220 to 280 g, kept adaptively for a week in a laboratory animal house. Tests were conducted in the same room, at 21–23 °C, in a similar manner for all animals; firstly, supine, and later, the prone position with tetrapodal fixation. Animals (N = 42) were divided into four groups, depending on the different options of the induced brain ischemia by occlusion of the common carotid artery. In group A (N = 10), stroke was induced by permanent clamping of the right carotid artery; in group B (N = 11), by permanent bilateral clamping of the carotid arteries; in group C (N = 10), by temporary clamping of the right carotid artery with its releasing; and in group D (N = 11), by temporary clamping the carotid arteries on both sides with their later releasing (Figure 1).

The outcome data were parameters of the amplitude (in µV) and latency (in ms) of the motor evoked potentials and the oral temperature (in °C) measured before and every minute at the subsequent stages of experiments. In 4 out of 46 animals (control group), stroke had not been induced but MEP and temperature measurements were performed in order to obtain reference values for comparison to occlusion effects and to parameters of other authors [21]. The total observation time in groups A and B was 10 min and 11 min in groups C and D. In animals from groups C and D, after five minutes of temporary occlusion of the common carotid artery on the right side or bilaterally, the “0 period” was applied for about one minute for the recovery of the arterial blood flow, including the elastic properties of the vessel after clamp release. The attempts to measure the blood pressure and PCO_2_ with a small cannula from the vein were abandoned because they might influence the arterial flow, an essential factor of stroke induction. After experiments, the brains were collected for cutting onto slides for histopathological examination. Histological verification also confirmed the influence of transcranial magnetic stimulation on the brain tissue structure in animals from the control group. The MEP parameters and the results from histological brain slides were read out independently by three observers, then discussed with and verified to obtain the final conclusions.

The study was conducted in accordance with the Declaration of Helsinki. Ethical considerations were also in agreement with Directive 2010/63/EU of the European Parliament and of the Council of 22 September 2010, on the protection of animals used for scientific purposes. Approval was also received from the Bioethical Committee of the University of Medical Sciences in Poznań (Poland), including studies on animals (decision no. 4/2008).

### 2.2. Anesthesia and Surgical Procedures

The animals were anesthetized intraperitoneally with ketamine hydrochloride (Ketanest, Park-Davis, Germany) at a dose of 90 mg/kg body weight. The depth of anesthesia was verified by a pupil dilatation and no toe pinch reflex could be elicited after approximately 25 min. The effect of the drug started approximately 20 min after administration and lasted for around 2 h. Small doses of ketamine (10 mg/kg) were added, when necessary, to maintain the anesthesia. To prevent aspiration of saliva into the lungs, the animals were premedicated by intramuscular injections with atropine sulfate (Polpharma, Poland) at a dose of 0.05 mg/kg.

When animals were placed dorsally, the common carotid artery was dissected uni- or bilaterally, 2 mm before it splits to internal and external carotid arteries. Dissected common carotid arteries were occluded and/or released depending on the group of tested animals (Figure 1).

When animals were placed ventrally, the right sciatic nerve was carefully dissected from the surrounding tissues before it splits into common peroneal and tibial nerves and prepared for recordings of evoked potentials. Special attention was paid not to dry the dissected nerve branches; they were soaked with drops of warm paraffin oil. The head of the animal was fixed with ear bars in a frame of the stereotactic apparatus while the upper and lower limbs were fixed to its base. The head of the animal was maintained at a distance of approximately 0.5 cm from the base. The procedure of anesthesia can cause the rats to lose their body temperature control, thus, the mounting base temperature was kept constant with a feedback-controlled heating blanket. The temperature of the animals was measured orally with an electronic probe; the temperature measured rectally could have been influenced by the heated base. Moreover, we considered a more proper measurement of the animal’s head area where the occlusion procedures were performed than the whole body heated from the base by the feedback system.

After completion of the experiments, a dose of ketamine two times greater than the lethal dose was administered intraperitoneally to animals. At the same time, the perfusion was performed (see Section 2.4). Next, a total craniotomy was performed, and brains were removed for histopathological procedures.

### 2.3. Electrophysiological Recordings

In order to determine changes in the efferent transmission of the corticospinal tract following the artery occlusions, transcranial magnetic stimulation (TMS) was applied to the contralateral motor cortex and MEP recordings were performed from the right sciatic nerve. The methodology was similar to the description presented by Kamida et al. [28] and our own experiences. After dissection of the sciatic nerve, it was placed on a bipolar, silver chloride recording electrode. The anode of the electrode pair was oriented closer to the spinal center while the cathode was distal. The distance between the poles of the recording electrodes was 3–4 mm. The ground electrode was placed on the muscle in close proximity to the recording electrodes.

TMS was performed with a 10 cm in diameter, circular magnetic coil (Medtronic A/S, Skøvlunde, Denmark) placed over the area of motor cortex for hindlimb innervation. The magnetic coil center (“hotspot”) was positioned from 1 to 5 mm lateral to the bregma. The coil was moved craniocaudally (±5 mm relative to the bregma with 1 mm steps) to optimize recorded MEP amplitudes (the largest) and latencies (the shortest). The coil was not angulated but placed flat onto the calvarial bone. The left side of the motor cortex was always stimulated with standard single pulses, assuming that the lateral corticospinal tract transmits efferent impulses mainly (in approximately 80%) by crossed projections of axons to the spinal cord motor centers of the opposite side. Hence, the optimal recording of MEPs was always performed from the right sciatic nerve. Moreover, we expected a small influence of the magnetic field generated by the coil on the right common carotid artery, which was most frequently occluded on the right side in all groups of animals. According to descriptions of Hovey and Jalinous [29], the coil with a diameter of 10–14 cm induces the greatest magnetic field strength at 660 V/m in the range from 2.2 to 2.7 T. In this study, the magnetic stimulation strength was defined as the percentage of maximal output. In most experiments on the rodents, this strength ranges from 30 to 70% [30]. This study used minimally invasive single magnetic stimuli, which means that their strength was adjusted to elicit MEP potentials with supramaximal amplitudes while minimizing movement artifacts for the stimulated objects that could affect the recording conditions. The right motor cortex close to the midline for innervation of hindlimb muscles was always stimulated with the B side of the coil. The analyzed MEP parameters were the amplitude, measured in µV, from the negative to positive deflection of the potential, and their latency, measured in ms, which is the time from the moment of motor cortex activation with a magnetic pulse is seen in the recording as the stimulus artifact to the potential’s onset. General rules for MEP recordings have been adopted for the standards described by Ferreira et al. [31], and the whole methodology on the TMS application in rats was very similar to the ones described by Nielsen et al. [32] and Linden et al. [33]. Motor evoked potentials were recorded using an integrated diagnostic system KeyPoint (Medtronic A/S, Skøvlunde, Denmark), configured with the R30 MagPro magnetic stimulator (Medtronic A/S, Skøvlunde, Denmark). All MEP recordings were made at 0.5 Hz with a low-pass filter setting while the upper-pass filter of KeyPoint was set at 2 kHz. The time base was set at 5 ms/D and the sensitivity of recording was from 0.5 to 10 mV/D.

### 2.4. Histological Verification of Ischemia

After completion of electrophysiological recordings, the animals were placed in a supine position on the operating table, the hearts were exposed, and perfusion was performed with physiological saline. After the blood flow was clarified, the skulls were carefully opened, brains removed, and stored in a 30% formalin solution for a period of one month. Horizontal, 90 μm thickness sections of brains were performed using freezing microtome with CO_2_ (Reichert, Austria). Cuts were performed sequentially at levels −2 mm, −3.1 mm, −4.1 mm, −5.1 mm from the bregma, placed on microscope slides, transferred through the histological reagents, stained with hematoxylin and eosin (H + E), and closed in DPX. Histological verification of the material obtained after staining procedures was performed in the bright field illumination microscopy with magnifications of 5× and 10× (Leitz, Germany). In most animals, the ischemic changes in brain tissue were due to changes either by direct necrotic ischemia or artifacts in the form of neural tissue defects resulting from cracks in small vessels. Locations of the changes in the microscopic image that were considered to be the result of ischemia were reconstructed on the transverse planes according to Paxinos and Watson’s [34] rat brain atlas and in the frontal planes according to Pellegrino et al. [35]. They were defined as the highest concentration in specific areas of the brain and their location was described according to the scheme of Garcia et al. [36].

### 2.5. Statistical Analysis

The results were analyzed using Statistica software STATISTICA v. 9.1 StatSoft (Kraków, Poland). Quantitative characteristics, such as amplitude, latency, and the temperature recorded in animals from groups A–D in every minute of the test, have been described by the mean value, standard deviation (SD), and the percentage of change with reference to the data obtained in a control group. Recordings of amplitudes of MEPs performed before occlusion of the common carotid artery were also treated as reference (control) values for each group of animals (100%). Wilcoxon test was used for comparisons of the amplitude and latency values from MEPs and the temperature at successive stages of observation (before and at subsequent minutes after arteries occlusion and/or releasing). The level of statistical significance was accepted at *p* < 0.05.

## 3. Results

The graphical presentation of the measured parameters variability is shown in the charts in Figure 2. A detailed summary of the results regarding the parameters of MEPs recorded in animals from groups A–D at various stages of observations are presented in Table 1.

There were no observed statistically significant changes between MEP parameters as well as temperatures recorded in animals from groups A–D after preparation of the surgical field at the neck prior to occlusion of the common carotid artery with the surgical clamp, and amplitudes (*p* = 0.07), latencies (*p* = 0.09), or temperatures (*p* = 0.08) measured in animals from the control group (Table 1). In four control animals, neither these results nor the histological verification confirmed the influence of transcranial magnetic stimulation on the brain tissue structure or the maneuvers during surgical preparation of the arteries to suspected MEP parameter fluctuations caused by the disturbed blood flow before occlusions.

In animals from groups A and B, results showed that five-minute uni- and bilateral occlusion of the common carotid arteries lead to alterations in brain blood circulation and evoked a significant decrease (at *p* = 0.06, see Table 2) in the MEP amplitudes by 19.1% and 23.1%, respectively (Table 1). Ten-minute occlusion in animals from groups A and B resulted in an MEP amplitude decrease (at *p* = 0.01 and *p* = 0.009) by 34.8% and 48.4%, respectively. In animals from groups C and D, after a similar course of the amplitude decrease during the first five minutes of occlusion by approximately 23.4% (at *p* = 0.04 and *p* = 0.03), the recovery in the arterial blood flow brought stabilization of the amplitude parameter at 11.9% and 23.0%, respectively, with reference to the initials values. The values of MEP amplitudes recorded in animals from groups C and D before occlusion did not differ significantly in the final observations after artery release (at *p* = 0.06 and *p* = 0.05, respectively).

The increase in MEP latency was by 0.9 ms on average (Table 1), reflecting the partial inability of corticospinal tract fibers to transmit neural impulses, which appeared to be significant only after the bilateral arteries occlusions in the fifth minute of recordings in animals from groups B and D (at *p* = 0.04, see Table 2). A significant change (at *p* = 0.03) in the average latency by 1.4 ms was observed in animals from group B in the final observation. However, the bilateral recovery of blood flow in animals from group D provided the subsequent shortening of this parameter in the last observation period, which did not differ significantly (at *p* = 0.05) from the initial value.

A similar course of the temperature change by 2.2 °C on average (Table 1) at *p* = 0.04 (Table 2) to the latency variability could be observed during the first five minutes of measurements in animals from groups B and D following bilateral occlusion. Permanent uni- or bilateral arterial ischemia in animals from groups A and B evoked a decrease in this parameter by up to 2.9 °C on average in the final observation (at *p* = 0.03 and *p* = 0.02, respectively). In both animals from groups C and D, a release of the common carotid arteries resulted in the recovery of temperature parameters with no significant differences to the values recorded before occlusions.

Examples of recordings in Figure 3 are convincing regarding the effects of arterial blood changes on the fluctuations of the MEPs amplitude parameter. Note the subsequential decrease in the MEPs amplitude parameter in A recorded following the permanent unilateral artery occlusion, and a different sequence of changes (increase after a decrease) shown in B, when the recordings were performed in an animal with bilateral, temporary occlusions.

In histological studies, results of ischemia were the most prominent bilaterally in sensory and motor areas mainly for the forelimb, rather than the hindlimb, innervation of the cortex, putamen and caudate nuclei, globulus pallidus, and areas adjacent to the fornix of the third ventricle (Figure 4).

## 4. Discussion

In this study, we evaluated the parameters of transcranially evoked motor evoked potentials recorded from the hindlimb nerve branches in rats following 5 min of uni- or bilateral common carotid artery occlusion and its release, which did not reveal total inhibition but reversible changes in the activity of corticospinal tract. Data presented in the graphs in parts B, D, and F of Figure 2 are convincing enough that the MEP amplitude parameter and the animal’s temperature, more than the MEP latency, are precise indicators of the reversible consequences of the ischemic stroke.

The neurophysiological method used in this study with the recording from nerves avoided the muscle stretch-related artifacts that could influence the precision of measurements of the MEP amplitudes and latencies, as well as the animals’ temperature. Moreover, such an approach diminished the possible risk of the applied anesthetics on the release of acetylcholine at the level of the neuromuscular junction, which could have been a source of false negative or positive results [30,37,38]. Thus, we have verified the hypotheses included in the study aims. Our previous experiences confirmed that MEPs recorded from the hindlimb nerves of rats following transvertebral stimulation is precise enough for evaluation of the regeneration process in the motor fibers of rat’s hindlimbs [30]. Moreover, MEPs evoked transcranially or transvertebrally are proven as a reliable and non-invasive diagnostic tool to assess the effects of stroke or peripheral nerve damage, and its subsequent spontaneous or therapeutically induced recovery [39,40,41,42,43]. A methodologically similar study by Kikonohana et al. [37] with transcranial motor evoked potentials revealed the high sensitivity of detection of the stroke effects after aortic occlusion in rats. In general, the morphology and parameters of the recorded MEPs before performing the ischemic procedures (see Table 1) is consistent with the description of the N1 and N2 components in the potentials recorded from the nerves by Shao et al. [21]. The MEP averaged amplitudes and latencies recorded in our study did not differ significantly from the parameters obtained in the works of Zhang et al. [38] and Fishback et al. [44], which were reported as 11.47 mV ± 5.25 mV and 5.1 ms ± 1.8 ms, respectively. The recorded MEPs main potential (see Figure 3), similar to the study of Linden et al. [33], reached the threshold amplitude at 15% of the stimulator output, while its maximal value was recorded at 60%.

A study by Konrad et al. [45] showed that MEPs morphology is altered more by pathological changes at the synaptic level than along the axons. Therefore, it should be assumed that in our studies, the expected changes after unilateral carotid artery clamping will show more changes at the level of the supraspinal centers of the brain than at the level of the spinal cord itself. According to Dougherty et al. [23] and Garcia et al. [24], unilateral or bilateral common carotid artery occlusion is an excellent experimental model of the effects of ischemic stroke. The results of our study seem to contradict the views of Longa et al. [27] regarding the lack of results of such a model due to numerous anastomoses compensating for the incorrect blood supply to the brain in critical conditions, and Kety et al. [26] that experimental blockade of the common carotid arteries does not cause significant changes in cerebral circulation in experimental conditions in animals and in clinical conditions in humans. In this study, we observed the effects of changes in cerebral circulation up to 10 min after the application of the effects of ischemia induced after temporary occlusion of the common carotid artery. Previous reports in related methodology included mainly long-term observations of the consequences of middle cerebral artery occlusion due to the frequency of stroke with this location, which is significant in the population of post-stroke patients, and therefore, worth a closer examination. In a study by Simpson and Baskin [20], MEPs that were recorded every hour for 6 h after occlusion of the middle cerebral artery showed significantly attenuated amplitudes. However, after about 5 h, the early latency components exceeded control values. These results indicate moderate reliability of the results of long-term ischemia in light of the long-term observation experimental studies.

The markings presented in Figure 4 show that the largest number of lesions resulting from ischemia at all examined levels of the brain from the bregma point were observed in animals from groups B and D, in which bilateral arterial occlusion was performed, also with the variant of restoring their flow. In animals with an ischemia model of occlusion of the right common carotid artery (groups A and C), artifacts were less numerous. It can be concluded that in animals of groups B and D, the markings were most often present unilaterally in the area of sensory and motor innervation of the limbs, mainly the anterior rather than the posterior cerebral cortex, the area of the putamen and the caudate nucleus bilaterally, the area of the globus pallidus bilaterally, and the area adjacent to the fornix of the third ventricle bilaterally. It should be noted that such changes illustrate the effects of short-term ischemia in a short period of observation. Nevertheless, they are structurally closely related to functional changes in the transmission of the corticospinal tract, where changes in the MEPs amplitude parameter were also most often noted and most clearly marked in the tested animals from groups B and D.

The purpose of histological verification in this study was to confirm the consistency of the results with the reports of other authors of experiments on rats after occlusion of the common carotid artery, concerning the effects of ischemia in the brain. Our study was not intended to quantify the effects of ischemia in detail, but to specify their most common locations. In addition, as can be seen from the slices in Figure 4, a similar pattern of unilateral (A and C) or bilateral (B and D) occlusion effects can be observed, whether secondary releasing has occurred or not. Similar to the work of Mathew et al. [46], using H + E staining of brain sections, this study defined the aftermath of stroke in the brain tissue as clear structural defects or empty areas with stained edges, mainly due to vascular rupture. There were no signs of bleeding in the lesion areas. Most authors of other reports described ischemic changes as structural discolorations or lesions resulting from the rupture of small vessels, similar to the observations reported in this paper. Nagasawa and Kogure [47] quantified artifacts according to the number of ventricles, discolorations, and brain tissue defects after H + E staining, most often in areas of the anterior neocortex after carotid artery clamping. This location closely coincides with our observations after histological verification of the brains of group B and D animals. Similar results of the neuroimaging and histochemical studies following the stroke caused in rats by the common carotid artery infarctions have been presented by Carmichael in his review [48].

In light of the presented literature data, which are not fully consistent with the observations in this work, it can be concluded that the verification of the results of MEP tests on the described model of a stroke may provide interesting data regarding the critical time of carotid artery occlusion, significantly affecting the activity of the motor cortex and the efferent neural transmission of the corticospinal pathways. The change in cerebral vascularity is probably compensated by the mechanism of arterial pressure compensation through anastomoses. Seitz et al. [49] showed that compensation for the effects of ischemia or general changes in arterial blood flow in the brain is affected by anastomoses and microanastomoses occurring between the arteries, which expand, thus eliminating blood flow disorders. This phenomenon seems to be one of the safety mechanisms allowing reorganization of the human cerebral circulation following cerebral ischemia induced by a change in arterial pressure. Confirmation of this phenomenon is a completely different course of stroke accompanying bilateral carotid artery occlusion, in which a sharp decrease in the MEP amplitude was observed, without the possibility of secondary blood supply compensation after bilateral carotid artery release within 5 min. The MEP amplitudes recorded in these cases were always significantly reduced throughout the duration of the ischemic model used and their values accounted for approximately 40% of the initial value.

Considering the limitations of the presented study, one should remember that the MEP study method does not fully specify the pathological phenomena occurring within the neurons of the motor cortex as a result of temporary, unilateral, or bilateral inhibition of blood flow within the carotid arteries, such as in episodes often described in clinical practice [50], indicating only model similarities in the ischemic stroke consequences. Moreover, our study was conducted on young adult male Wistar rats whereas, clinically, stroke affects mostly aged and comorbid, including obesity, patients [51]. Taking into account the differences in the ability of the rodent brain and the human brain to withstand ischemic consequences, such as stroke, it is important to note that this study was conducted using rats as subjects and further research is needed to determine whether these findings are applicable to humans.

A contemporary clinical observational study hypothesized that previous transient ischemic attack (TIA) had a favorable effect on early outcomes after acute nonlacunar ischemic stroke [52]. Prior TIA was associated with a favorable outcome in nonlacunar stroke, suggesting its neuroprotective effect possibly by inducing a phenomenon of ischemic tolerance, allowing better recovery from a subsequent ischemic stroke. The proposed model used in our study with the assessment of the corticospinal system function could also be used in the future for the evaluation of the transient ischemic attack effects under experimental conditions. Moreover, the future line of research on the discussed topic would be to study the inhibition of the corticospinal tract neuronal activity in experimental conditions in cases of acute small vessel disease versus other stroke subtypes. This recommendation seems to be reasonable because the pathophysiology, prognosis, and clinical features of small vessel acute ischemic stroke are different from other stroke subtypes [53].

## 5. Conclusions

The parameters of transcranially evoked motor evoked potentials recorded from the hindlimb nerve branches in rats following 5 min of uni- or bilateral common carotid artery occlusion and its release did not reveal total inhibition but instead reversible changes in the activity of corticospinal tract. A five-minute occlusion of the common carotid artery on one or both sides causes a decrease in body temperature by approximately 1.5 °C, while their releasing after 5 min of observation causes compensation of this parameter to a statistically comparable value before the ischemia was caused. The parameters of MEP amplitude and temperature variability are more sensitive in monitoring the efferent transmission of the corticospinal tract than the MEP latency parameter in observing the course of ischemic effects in experimental conditions. Results of ischemia in histological studies were the most prominent bilaterally in sensory and motor areas mainly for the forelimb, rather than hindlimb, innervation of the cortex, putamen and caudate nuclei, globulus pallidus, and areas adjacent to the fornix of the third ventricle. The symptoms of rat brain infarction are much more optimistic than those described in patients after stroke and require further comparison with the clinical observations.

## Figures and Tables

**Figure 1 biomedicines-11-01287-f001:**
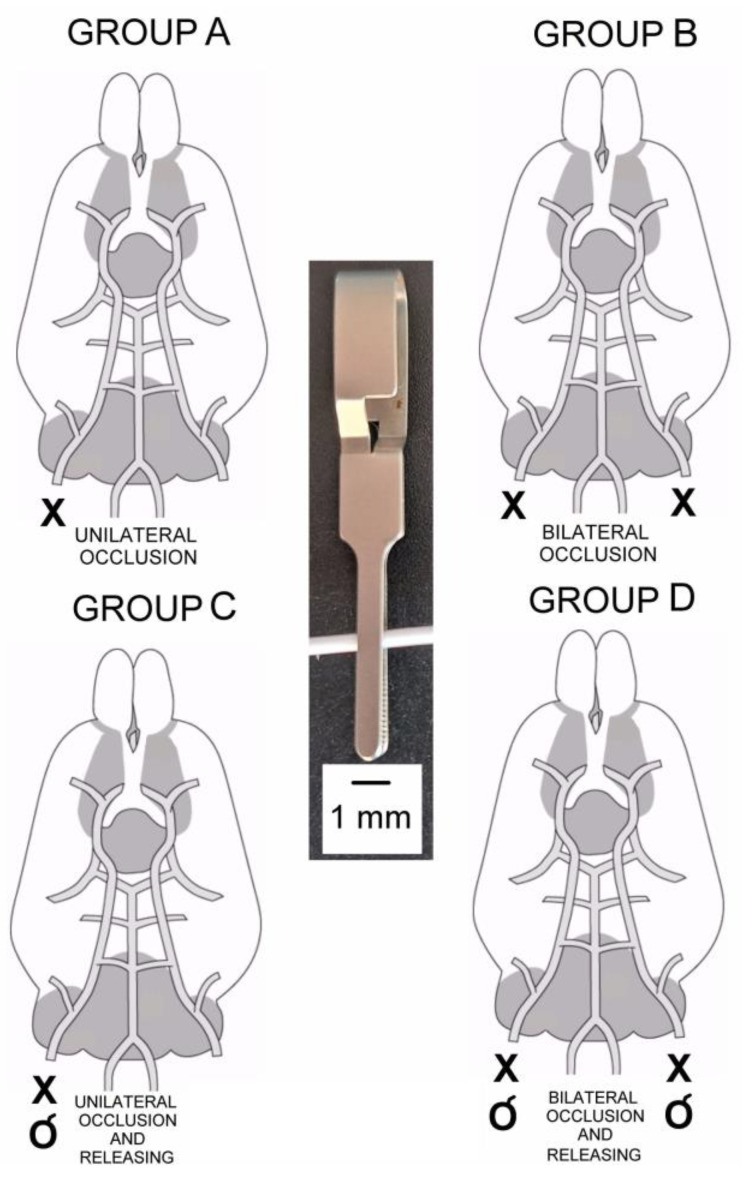
Diagrams of four types of induced ischemic stroke on plans of the rat brain vasculature. Group A—permanent unilateral occlusion of the common carotid artery (always right), group B—permanent bilateral occlusion of the common carotid arteries, group C—temporary unilateral occlusion of the common carotid artery (always right) with its releasing, group D—temporary bilateral occlusion of the common carotid arteries with their release. The “X” means occlusion, while “O” means the artery release. The center of the figure shows a surgical clamp used to occlude the common carotid artery.

**Figure 2 biomedicines-11-01287-f002:**
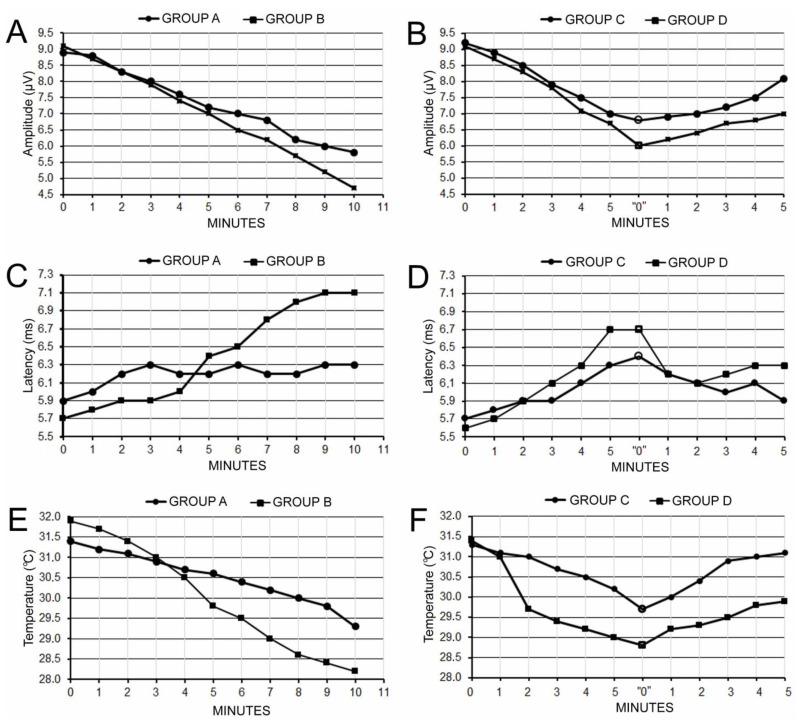
Variability of the amplitude (**A**,**B**) and latency (**C**,**D**) of the MEPs, and temperature (**E**,**F**) recorded in rats from groups A and B (**left side**) and C and D (**right side**) at various stages of observations. “0”—releasing the artery after five minutes of occlusion.

**Figure 3 biomedicines-11-01287-f003:**
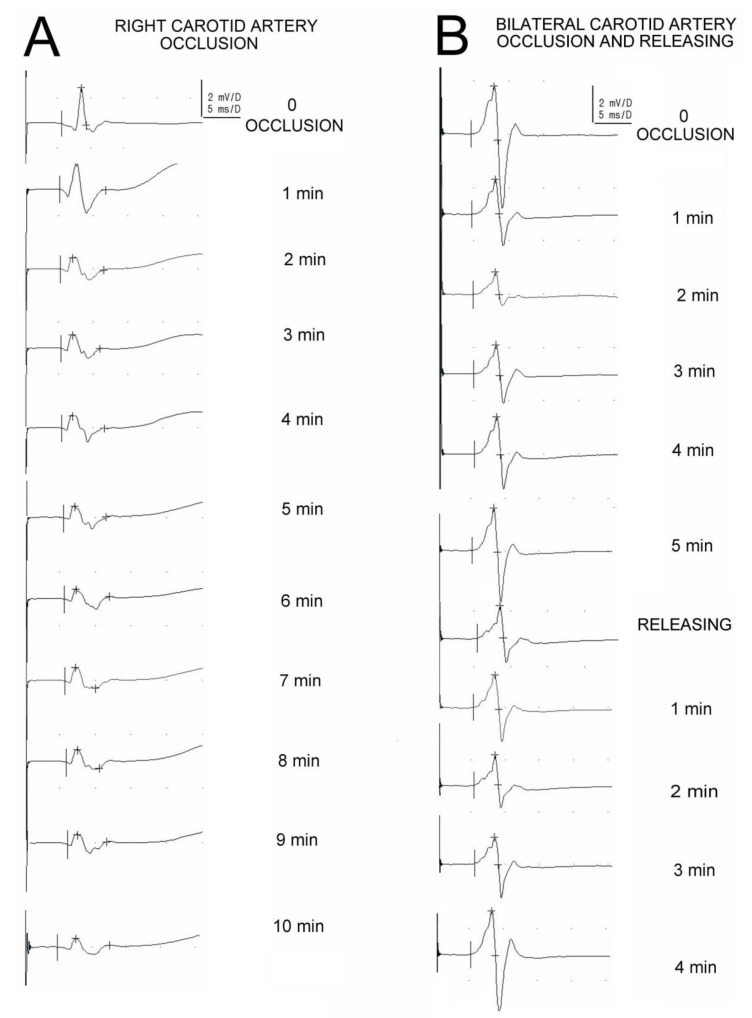
Examples of motor evoked potential recordings with the observation time of up to 10 min (**A**) in one of the animals from group A as a result of the right carotid artery occlusion, and (**B**) in one of the animals from group D with the bilateral artery occlusion and release.

**Figure 4 biomedicines-11-01287-f004:**
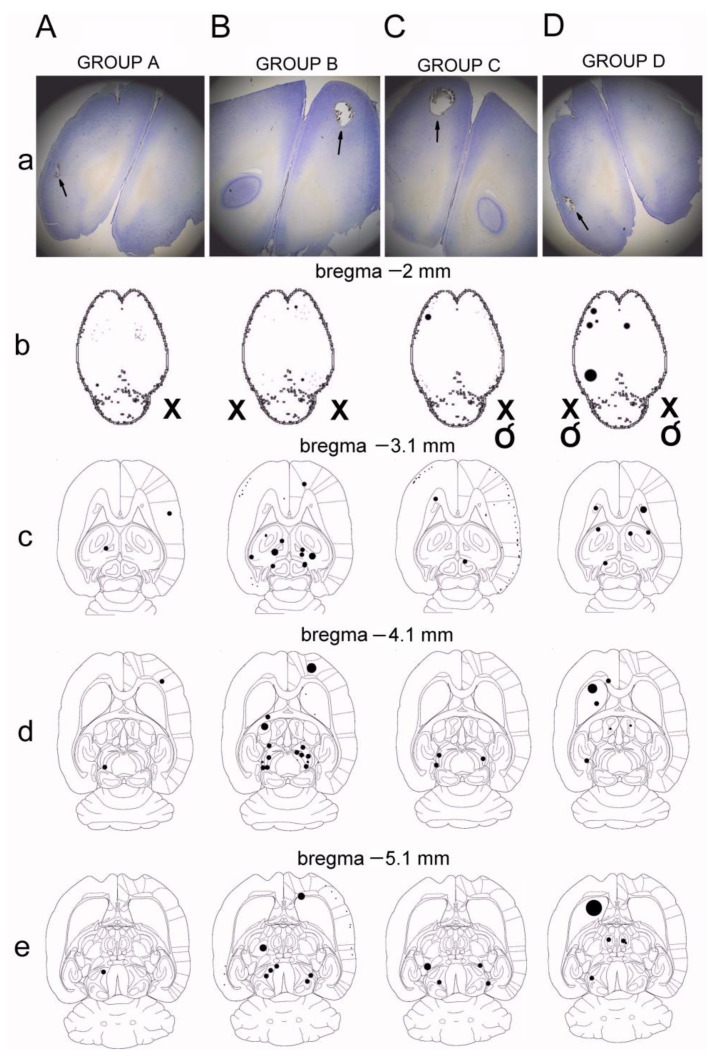
Locations of lesions resulting from the induction of ischemia with various variants found in animals from A–D groups (“X” means occlusion, while “O” is the artery releasing). Examples of microscopic images with the lesion sites marked with arrows in the four groups of animals (**A**–**D**) are shown in part “(**a**)” of the figure. The results of histological examinations (**b**–**d**) were re-drawn from the microscopic images and showed the location of lesions of varying severity, which are marked with circles of different diameters on the cross-sectional diagrams of the brain. In “(**b**)”, cross-sections are shown at the levels −2 mm from the bregma point, in “(**c**)“, −3.1 mm, in “(**d**)”, −4.1 mm, and in “(**e**)”, −5.1 mm. Black arrows indicate ischemic spots.

**Table 1 biomedicines-11-01287-t001:** Summary of mean values and standard deviations of motor evoked potential amplitudes (expressed in percentages) and latencies, as well as temperature values recorded in four groups of animals at different stages of observation. Data on the analogical parameters measured in the control group of animals with no surgeries are presented on the left side of the table.

		**Amplitude (µV), % of Change**											
		**Duration of Ischemia after Carotid Artery Occlusion (min)**											
Control	Animals Groups	Before	1	2	3	4	5	6	7	8	9	10	
9.2 ± 1.3 (N = 4)	A (N = 10)	8.9 ± 1.1	8.8 ± 1.0	8.3 ± 0.9	8.0 ± 1.1	7.6 ± 0.9	7.2 ± 1.2	7.0 ± 1.1	6.8 ± 0.9	6.2 ± 1.0	6.0 ± 1.2	5.8 ± 1.1	
		−100%	−1.20%	−6.80%	−10.10%	−14.60%	−19.10%	−20.30%	−23.60%	−30.30%	−32.60%	−34.80%	
	B (N = 11)	9.1 ± 1.3	8.7 ± 1.0	8.3 ± 0.7	7.9 ± 0.8	7.4 ± 0.9	7.0 ± 1.0	6.5 ± 1.1	6.2 ± 0.9	5.7 ± 1.1	5.2 ± 1.0	4.7 ± 0.9	
		−100%	−4.40%	−8.80%	−13.20%	−18.70%	−23.10%	−28.60%	−31.90%	−37.40%	−42.90%	−48.40%	
		Occlusion duration (min)							Occlusion releasing duration (min)				
	Animals Groups	Before	1	2	3	4	5	“0”	1	2	3	4	5
	C (N = 10)	9.2 ± 1.2	8.9 ± 1.0	8.5 ± 1.0	7.9 ± 1.0	7.5 ± 0.8	7.0 ± 1.0	6.8 ± 0.9	6.9 ± 1.0	7.0 ± 0.9	7.2 ± 1.2	7.5 ± 1.1	8.1 ± 1.1
		−100%	−3.30%	−7.60%	−14.10%	−18.50%	−23.90%	−26.10%	−25.00%	−23.90%	−21.70%	−18.40%	−11.90%
	D (N = 11)	9.1 ± 1.3	8.7 ± 0.9	8.3 ± 0.9	7.8 ± 1.0	7.1 ± 1.0	6.7 ± 0.7	6.0 ± 1.1	6.2 ± 1.1	6.4 ± 0.8	6.7 ± 3.1	6.8 ± 0.8	7.0 ± 0.9
		−100%	−4.40%	−8.80%	−14.30%	−21.90%	−26.30%	−34.10%	−31.80%	−10.90%	−11.00%	−25.20%	−23.00%
		**Latency (ms)**											
5.8 ± 0.8 (N = 4)		**Duration of Ischemia After Carotid Artery Occlusion (min)**											
	Animals Groups	Before	1	2	3	4	5	6	7	8	9	10	
	A (N = 10)	5.9 ± 0.8	6.0 ± 0.8	6.2 ± 0.6	6.3 ± 0.9	6.2 ± 0.5	6.2 ± 0.9	6.3 ± 0.7	6.2 ± 0.7	6.2 ± 0.7	6.3 ± 0.7	6.3 ± 0.8	
	B (N = 11)	5.7 ± 0.7	5.8 ± 0.7	5.9 ± 0.8	5.9 ± 0.7	6.0 ± 0.7	6.4 ± 0.5	6.5 ± 0.8	6.8 ± 0.8	7.0 ± 0.9	7.1 ± 0.9	7.1 ± 0.8	
		Occlusion duration (min)						Occlusion releasing duration (min)					
	Animals Groups	Before	1	2	3	4	5	“0”	1	2	3	4	5
	C (N = 10)	5.7 ± 0.4	5.8 ± 0.8	5.9 ± 0.6	5.9 ± 0.4	6.1 ± 0.4	6.3 ± 0.4	6.4 ± 0.4	6.2 ± 0.4	6.1 ± 0.5	6.0 ± 0.5	6.1 ± 0.4	5.9 ± 0.6
	D (N = 11)	5.6 ± 0.7	5.7 ± 0.5	5.9 ± 0.6	6.1 ± 0.6	6.3 ± 0.6	6.7 ± 0.6	6.7 ± 0.5	6.2 ± 0.5	6.1 ± 0.5	6.2 ± 0.4	6.3 ± 0.4	6.3 ± 0.5
		**Temperature (°C)**											
31.6 ± 1.1 (N = 4)		**Duration of Ischemia after Carotid Artery Occlusion (min)**											
	Animals Groups	Before	1	2	3	4	5	6	7	8	9	10	
	A (N = 10)	31.4 ± 1.0	31.2 ± 1.2	31.1 ± 1.2	30.9 ± 1.2	30.7 ± 1.1	30.6 ± 1.1	30.4 ± 0.9	30.2 ± 1.1	30.0 ± 1.2	29.8 ± 1.1	29.3 ± 1.1	
	B (N = 11)	31.9 ± 1.2	31.7 ± 1.3	31.4 ± 1.2	31.0 ± 1.3	30.5 ± 0.9	29.8 ± 1.2	29.5 ± 0.9	29.0 ± 1.1	28.6 ± 1.0	28.4 ± 1.2	28.2 ± 1.2	
		Occlusion duration (min)							Occlusion releasing duration (min)				
		Before	1	2	3	4	5	“0”	1	2	3	4	5
	C (N = 10)	31.3 ± 1.2	31.1 ± 1.2	31.0 ± 1.0	30.7 ± 0.9	30.5 ± 1.0	30.2 ± 1.0	29.7 ± 1.2	30.0 ± 1.1	30.4 ± 1.0	30.9 ± 0.9	31.0 ± 1.1	31.1 ± 1.0
	D (N = 11)	31.4 ± 0.9	31.0 ± 0.9	29.7 ± 1.0	29.4 ± 1.0	29.2 ± 1.0	29.0 ± 1.1	28.8 ± 0.9	29.2 ± 1.0	29.3 ± 0.9	29.5 ± 0.9	29.8 ± 0.9	29.9 ± 0.9

Abbreviations: Groups of animals A—permanent unilateral occlusion of the common carotid artery (always right), group B—permanent bilateral occlusion of the common carotid arteries, group C—temporary unilateral occlusion of the common carotid artery (always right) with its releasing, group D—temporary bilateral occlusion of the common carotid arteries with their releasing; “0”—releasing the artery after five minutes of occlusion.

**Table 2 biomedicines-11-01287-t002:** Differences (*p*) in parameters of MEPs and the temperature recorded at different periods of observations in the four groups of animals.

Observation Periods (Minutes)	Group A			Group B			Group C			Group D		
	Amplitude	Latency	Temperature	Amplitude	Latency	Temperature	Amplitude	Latency	Temperature	Amplitude	Latency	Temperature
“0” vs. 5th	**0.06**	0.05	0.05	**0.06**	**0.04**	**0.04**	**0.04**	0.05	0.05	**0.03**	**0.04**	**0.04**
“0” vs. 10th	**0.01**	0.05	**0.03**	**0.009**	**0.03**	**0.02**	0.05	0.07	0.07	**0.04**	0.05	0.05

Abbreviation: “0”—artery releasing; *p* < 0.05 significant differences are marked in bold.

## Data Availability

All data described in this study are presented in the manuscript.

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
