# Peer review of "Temporary Occlusion of Common Carotid Arteries Does Not Evoke Total Inhibition in the Activity of Corticospinal Tract Neurons in Experimental Conditions"

_biomedicines, 2023, doi:10.3390/biomedicines11051287_

Round 1
Reviewer 1 Report
The authors present the results of an experimental study conducted with 46 adult male Wistar rats induced with four types of induced ischemic stroke: Group A – permanent unilateral occlusion of the common carotid artery, group B - permanent bilateral occlusion of the common carotid arteries, group C - temporary unilateral occlusion of the common carotid artery with its releasing, group D - temporary bilateral occlusions of the common carotid arteries with its releasing. The outcome data were the parameters of amplitudes and latencies of the motor evoked potentials and the oral temperature. Histological verification of the results of ischemia were analyzed The authors found that the parameters of transcranially motor evoked potentials recorded in the hindlimb nerve branches of rats after 5 minutes duration of the uni- or bilateral common carotid artery occlusion and release did not reveal total inhibition, but the reversible changes in the activity of corticospinal tract, concluding that the symptoms of rat's brain infarction are much more optimistic than those described in patients after stroke and require further comparison with the clinical observations. The study is potentially interesting, but can improved if the following considerations are addressed:
- It would be helpful to mention that the previous presence of a TIA is associated -in humans- with a good early outcome in non-lacunar ischemic strokes, thus suggesting a neuroprotective effect of TIA possibly by inducing a phenomenon of ischemic tolerance (see and add this reference Cerebrovasc Dis 2004; 18. 304-311). Did the authors consider this in their experimental study?
- It would be interesting if the authors included in the text some of the limitations of this study.
- It would be interesting to emphasize in the text that a future line of research on the discussed topic would be to study the Inhibition of the Corticospinal Tract Neuron Activity in Experimental Conditions in acute small vessel disease versus other stroke subtypes. This recommendation is because the pathophysiology, prognosis, and clinical features of small vessel acute ischemic strokes are different from other stroke subtypes (Int J Mol Sci 2022; 23, 1497).
Reviewer 2 Report
In this manuscript, the authors investigated the effects of temporary occlusion of carotid arteries and releasing after 5 mins in rats. They recorded transcranially evoked motor evoked potentials from hindlimb nerve branches, and found that temporary occlusion of common carotid arteries does not evoke total inhibition in the activity of corticospinal tract neurons. This suggests that such occlusion may not have as severe neurological consequences as previously thought. Overall, this research highlights the effects of temporary occlusion on neural activity and body temperature, although the differences shown in the results were not very significant. However, due to the scarcity of similar studies, I think this manuscript still provides some meaningful information, which may be useful in the field of ischemia stroke in the future. Some of my minor comments are also attached below.
1. Methods: I don't really understand the narrative part of Electrophysiological recordings. The authors described "MEP recordings were performed from the right sciatic nerve...". Since carotid arteries are ligated on the right side, it is theoretically more reasonable to record signals on the opposite side (left side)?
2. Tab. 1: The procedure of anesthesia can cause the rats to lose their body temperature control (hyperthermia), so a heating plate is required to help keep them warm. The body temperature of such an animal mostly depends on the strength of the external warmer, so is it meaningful to measure the body temperature of the rat at this time? In addition, the oral temperature is relatively easy to be disturbed by the outside surroundings (and the probe containing metal components will be very close to the TMS coil). Why did the authors not use rectal temperature measurement, which is generally more commonly used and closer to core body temperature?
3. Tab. 2: This table does not seem necessary. Authors can consider directly marking in Fig. 2 with different symbols, which I think will be more conducive to readers’ interpretation.
4. Fig. 3A: Why did the wave pattern of the amplitude not return to the ground state after the stimulus ended, but showed a slight upward trend? Does this situation also occur in the bilateral permanent occlusion group? I think the authors should provide more explanation for this.
5. Fig. 4: Group C does not seem to be injured as slightly as Group A (even looks slightly more serious), why? In fact, I think these results should be seriously considered to be quantified, otherwise the above-mentioned ambiguity will appear.
6. Taking into account the differences in the ability of the rodent brain and the human brain to withstand injury, it is important to note that this study was conducted using rats as subjects, and further research is needed to determine whether these findings are applicable to humans. I think the authors should emphasize this. In addition, I would also suggest that the authors consider putting forward a prospective view of the possible clinical application of their results (especially TIA).
Some minor typo errors in the manuscript should be corrected.
Reviewer 3 Report
This study addessed the neural efferent transmission within fibers of the corticospinal tract in in a rat model of ischemia. Efferent transmission of the corticospinal tract was verified by 22 motor evoked potential (MEP) recordings from the sciatic nerve after transcranial magnetic stimulation. MEPs amplitude and latencies parameters, oral measurements of temperature, and verification of ischemic effects in brain slides stained with hematoxylin and eosin staining (H+E) were analyzed. They concluded that the parameters of transcranially evoked motor evoked potentials recorded from the hindlimb nerve branches in rats following 5 minutes lasting the uni- or bilateral common carotid artery occlusion and its releasing did not reveal the total inhibition but the reversible changes in the activity of corticospinal tract. This is an interesting neurophysiological study. However, the study was conducted on young adult male Wistar rats whereas, clinically, stroke affects mostly aged, comorbid, including obesity, patients (see, DOI: 10.1111/ACEL.12678)
The quality of the English language is acceptable
